# Gut Microbiota Serves a Predictable Outcome of Short-Term Low-Carbohydrate Diet (LCD) Intervention for Patients with Obesity

Susu Zhang,[a,b] Peili Wu,[a,d] Ye Tian,[a,b] Bingdong Liu,[b,c] Liujing Huang,[b] Zhihong Liu,[b] Nie Lin,[a,e] Ningning Xu,[a] Yuting Ruan,[a] Zhen Zhang,[a] Ming Wang,[f] Zongbing Cui,[b] HongWei Zhou,[g] Liwei Xie,[a,b,h] Hong Chen,[a] Jia Sun[a]

[a]Department of Endocrinology and Metabolism, Zhujiang Hospital, Southern Medical University, Guangzhou, China

[b]State Key Laboratory of Applied Microbiology Southern China, Guangdong Provincial Key Laboratory of Microbial Culture Collection and Application, Guangdong Open Laboratory of Applied Microbiology, Institute of Microbiology, Guangdong Academy of Sciences, Guangzhou, China

[c]The First Affiliated Hospital of Jinan University, Guangzhou, China

[d]Department of Endocrinology and Metabolism, Nanfang Hospital, Southern Medical University, Guangzhou, China

[e]Department of Endocrinology and Metabolism, Shantou Central Hospital, Shantou, China

[f]Nephrology Center of Integrated Traditional Chinese and Western Medicine, Zhujiang Hospital, Southern Medical University, Guangzhou, China

[g]State Key Laboratory of Organ Failure Research, Microbiome Medicine Center, Division of Laboratory Medicine, Zhujiang Hospital, Southern Medical University, Guangzhou, China

[h]School of Public Health, Xinxiang Medical University, Xinxiang, China

Susu Zhang, Peili Wu, Ye Tian, and Bingdong Liu contributed equally to this article. The author order was determined by their equal but gradated contributions for this paper.

**ABSTRACT** To date, much progress has been made in dietary therapy for obese patients. A low-carbohydrate diet (LCD) has reached a revival in its clinical use during the past decade with undefined mechanisms and debatable efficacy. The gut microbiota has been suggested to promote energy harvesting. Here, we propose that the gut microbiota contributes to the inconsistent outcome under an LCD. To test this hypothesis, patients with obesity or patients who were overweight were randomly assigned to a normal diet (ND) or an LCD group with *ad libitum* energy intake for 12 weeks. Using matched sampling, the microbiome profile at baseline and end stage was examined. The relative abundance of butyrate-producing bacteria, including *Porphyromonadaceae Parabacteroides* and *Ruminococcaceae Oscillospira*, was markedly increased after LCD intervention for 12 weeks. Moreover, within the LCD group, participants with a higher relative abundance of *Bacteroidaceae Bacteroides* at baseline exhibited a better response to LCD intervention and achieved greater weight loss outcomes. Nevertheless, the adoption of an artificial neural network (ANN)-based prediction model greatly surpasses a general linear model in predicting weight loss outcomes after LCD intervention. Therefore, the gut microbiota served as a positive outcome predictor and has the potential to predict weight loss outcomes after short-term LCD intervention. Gut microbiota may help to guide the clinical application of short-term LCD intervention to develop effective weight loss strategies. (This study has been registered at the China Clinical Trial Registry under approval no. ChiCTR1800015156).

**IMPORTANCE** Obesity and its related complications pose a serious threat to human health. Short-term low-carbohydrate diet (LCD) intervention without calorie restriction has a significant weight loss effect for overweight/obese people. Furthermore, the relative abundance of *Bacteroidaceae Bacteroides* is a positive outcome predictor of individual weight loss after short-term LCD intervention. Moreover, leveraging on these distinct gut microbial structures at baseline, we have established a prediction

Address correspondence to Liwei Xie, xielw@gdim.cn, Hong Chen, chenhong123@smu.edu.cn, or Jia Sun, sunjia@smu.edu.cn.

model based on the artificial neural network (ANN) algorithm that could be used to estimate weight loss potential before each clinical trial (with Chinese patent number 2021104655623). This will help to guide the clinical application of short-term LCD intervention to improve weight loss strategies.

**KEYWORDS** gut microbiota, low-carbohydrate diets, *Bacteroidaceae Bacteroides*, artificial neural network, obesity, weight loss

Obesity or obesity-related chronic diseases affect over 2 billion people worldwide (1). Obesity is a chronic metabolic disease caused by multiple factors, including, but not limited to, consumption of inexpensive and calorie-dense foods, decreased physical activity, insulin resistance, or psychosocial factors (2). Indeed, epidemiologic data from the National Center for Health Statistics (NCHS) show that the age-adjusted prevalence of obesity in adults was 42.4% in 2017 to 2018 in the United States (3). Likewise, in China, more than half of Chinese adults are overweight or obese, according to the Report on Chinese Residents' Chronic Diseases and Nutrition in 2020 (http://www.gov.cn/xinwen/2020-12/24/content_5572983.htm). Meanwhile, obesity is still a detrimental factor for a plethora of chronic diseases, such as cardiovascular diseases (CVDs), diabetes, and cancer, which has an adverse impact on overall health (4). Body mass index (BMI)-related CVDs account for 41% of deaths and 34% of disabilities and is the leading cause of adverse events (5).

The substantially increased epidemic (3), latent health hazards, and huge medical expenditures (6) of obesity require the identification of effective intervention strategies. For instance, lifestyle interventions, obesity pharmacotherapy, and bariatric surgery have been proven and granted by the Guideline Recommendations for Obesity Management (7). Lifestyle interventions for weight loss are the cornerstone for obesity treatment (8). Among all lifestyle interventions, dietary intervention is the optimal choice for promoting weight loss. A large number of different dietary approaches, such as low-carbohydrate diet (LCD), high-protein diets, low-fat diets, low-glycemic index diets, balanced-deficit diets, and vegetarian-, vegan-, and Mediterranean-style diets, have been reported (7). These eating patterns with various macronutrient distributions have substantial/spurious benefits in certain groups of patients (7, 8). As a result, guidelines lack a consensus regarding the best dietary type to produce weight loss (7, 9). Dietary carbohydrate restriction for obesity treatment has attracted public attention in recent years, although the ratios of macronutrients to LCD have not yet been standardized (10, 11). Proponents claimed that carbohydrate restriction is closely associated with decreased plasma insulin levels, followed by elevated fat oxidation, energy expenditure and weight loss (10). Thus, LCD is an effective and feasible weight loss strategy, especially for those with obesity-related chronic diseases (e.g., type 2 diabetes) (12). However, in other clinical investigations, weight loss after LCD intervention was not significantly changed (9, 10). Therefore, the efficacy of LCD interventions for weight management are inconsistent across different studies.

Although weight loss associated with LCD intervention has been reported in different clinical trials, the exact benefit and sustainability remain a challenge to quantify. Nonetheless, lack of sufficient evidence to evaluate the heterogeneity regarding weight loss under LCD intervention (13) limits the application of LCD. To address this discrepancy, further confirmatory studies have provided insight into the gut microbiota in the gastrointestinal track in recent years (14, 15). The gut microbiota is essential in processing dietary polysaccharides and has a further impact on acquisition and storage of fat (14). Adjusting dietary patterns may alter the composition and diversity of gut microbiota (16). Animal experiments showed that germ-free (GF) mice gained body weight and presented obesity-relevant metabolic phenotypes after fecal transplantation from obese twins. However, these symptoms were reversed by cohousing with mice harboring the microbiota of the lean cotwin. This study revealed that specific members of *Bacteroidetes* from the microbiota of the lean cotwin account for diet-

dependent results (17). In addition to the increased abundance of *Bacteroides* and a reduction of *Firmicutes* (18), the body fat of GF mice was increased after transplantation of the gut microbiota from obese mice. Furthermore, some species of *Bacteroides* were reported to be more abundant in lean people (19). Additionally, a 3-day high-calorie diet intervention study (20) reported that the relative abundance of *Bacteroidetes* decreased in the human gut after dietary intervention. Fouladi et al. reported that bariatric surgery, such as Roux-en-Y gastric bypass (RYGB) surgery, significantly shifts the composition and abundance of gut microbiota, which could potentially contribute to weight loss and metabolic benefits (21). Together, these results implied that *Bacteroides* and *Firmicutes* may play diverse roles in the pathogenesis of obesity.

Therefore, the present study was undertaken to verify the hypothesis that inconsistent weight loss outcome under LCD intervention is due to the variation of gut microbiota composition. Our investigation confirmed that short-term (12-week) LCD intervention results in significant weight loss and elevation of certain groups of gut microbiota. Taking advantage of the advanced computation algorithms, such as random forest and artificial neural networks (ANNs), we identified that a higher relative abundance of *Bacteroidaceae Bacteroides* at baseline results in distinct weight loss outcomes under LCD intervention. From the current investigation, we demonstrated that the relative abundance of *Bacteroidaceae Bacteroides* is a positive outcome predictor of individual weight loss after LCD intervention. Moreover, leveraging on these distinct gut microbial structures at baseline, we have established a prediction model based on the ANN algorithm to estimate weight loss potential and efficacy for each clinical trial with the purpose to improve weight loss strategies.

## RESULTS

**Clinical characterizations of participants in the weight loss trial.** To assess the effect of LCD intervention on weight loss and explore potential unidentified biomarkers associated with weight loss efficacy, a total of 51 eligible overweight or obese participants were recruited in the present study. Their BMI and age ranged from 26.2 to 40.94 kg m$^{-2}$ and from 21 to 59 years old, respectively. The overall weight loss trial was divided into two stages (stage I, baseline stage; stage II, end stage), and an overview of the whole study is illustrated in Fig. 1A. Fifty-one participants were recruited and randomly assigned into two groups (the normal diet [ND] group [$n = 25$] and the LCD group [$n = 26$]). The clinical characteristics of participants at the baseline stage are summarized in Table 1. Baseline information about the study participants, including age, waist circumference, and visceral fat area (VFA), between the ND and LCD groups were not significantly different (35.80 $\pm$ 8.27 versus 36.58 $\pm$ 8.70 years, 90.98 $\pm$ 8.22 versus 94.28 $\pm$ 9.69 cm, and 104.90 $\pm$ 25.98 versus 122.00 $\pm$ 37.60 cm$^2$, respectively). Furthermore, there was no significant difference in glycometabolism, lipid metabolism, or hepatic and renal function between the two groups. However, the average BMI, waist-to-hip ratio (WHR), and body fat ratio (BFR) at baseline were higher in the LCD group than those in the ND group (BMI, 28.61 $\pm$ 2.04 versus 30.44 $\pm$ 3.38 kg m$^{-2}$; WHR, 0.87 $\pm$ 0.05 versus 0.90 $\pm$ 0.05; BFR [%], 33.18 $\pm$ 4.16 versus 36.53 $\pm$ 5.10, respectively). The weight loss trial lasted for 12 weeks with either ND or LCD intervention without energy restriction. No antibiotics or drugs were taken either 3 months before or during the course of this weight loss trial.

**Short-term LCD intervention results in obvious weight loss for obese/overweight participants.** LCD has a plethora of definitions, for example, carbohydrate reduction from 26 to 45% of total calories from the American Diabetes Association (ADA); but in the review by the National Lipid Association (NLA), 10 to 25% of total calories from carbohydrates for an LCD was adopted (22). In the present study, we adopted the NLA criterion, more specifically, a range of 10 to 25% of the total daily energy from carbohydrates (50 to 130 g day$^{-1}$) (23). Three-day 24-h dietary recalls, which are used by medical professionals, nutrition specialists, and social scientists (https://dietassessmentprimer.cancer.gov/profiles/recall/index.html; https://en.wikipedia.org/wiki/24-hour_diet_recall#cite_note-b-1), were provided by participants every week. We calculated the proportions of three

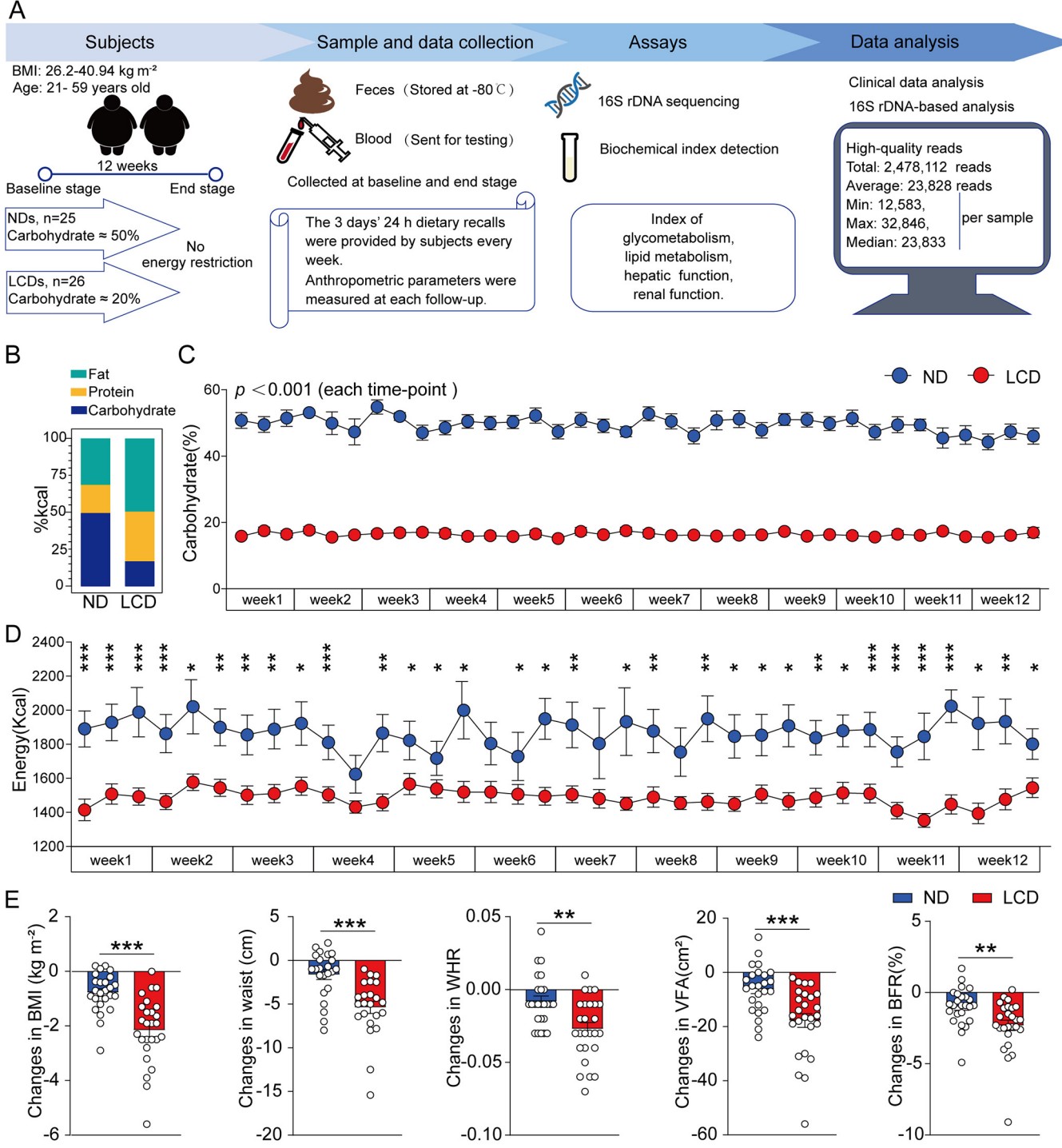

FIG 1 Overview of the study, dietary information, and variation of anthropometric parameters. (A) Schematic overview of the study design. (B) Daily diet composition of three macronutrients in the normal diet (ND) and low-carbohydrate diet (LCD) group, respectively, represented in percent calories. (C) Average proportions of carbohydrates were about 50% of the total in the ND group and up to 20% of the LCD group, calculated from 24-h dietary recalls of 3 days in every week. Data are expressed as mean ± SEM. P values are from unpaired, two-sided t tests. (D) Average energy intake calculated from food conversion was generally higher in the ND group than in the LCD group. Data are expressed as mean ± SEM, and values are from unpaired, two-sided Student's t test; *, $P < 0.05$; **, $P < 0.01$; ***, $P < 0.001$; NS, not significant. (E) Participants in the LCD group achieved a distinct decrease in BMI, waist circumference, WHR, BFR, and VFA compared to those values observed in the ND group at the end stage. Data are expressed as mean ± SEM, and values are from unpaired, two-sided Student's t test; **, $P < 0.01$; ***, $P < 0.001$.

**TABLE 1** Baseline clinical characteristics between the ND group and LCD group

| Parameters[a] | ND ($n$ = 25)[b] | LCD ($n$ = 26)[b] | $P$ value[c] |
|---|---|---|---|
| Female/male | 16/9 | 22/4 | |
| Age, yrs | 35.80 ($\pm$8.27) | 36.58 ($\pm$8.70) | 0.745 |
| BMI, kg m$^{-2}$ | 28.61 ($\pm$2.04) | 30.44 ($\pm$3.38) | 0.024[d] |
| BMR, kcal | 1,497.41 ($\pm$202.64) | 1,494.70 ($\pm$172.32) | 0.959 |
| Waist, cm | 90.98 ($\pm$8.22) | 94.28 ($\pm$9.69) | 0.196 |
| WHR, ratio | 0.87 ($\pm$0.05) | 0.90 ($\pm$0.05) | 0.046[d] |
| BFR, % | 33.18 ($\pm$4.16) | 36.53 ($\pm$5.10) | 0.013[d] |
| VFA, cm$^2$ | 104.90 ($\pm$25.98) | 122.00 ($\pm$37.60) | 0.056 |
| LBM, kg | 50.17 ($\pm$9.97) | 50.38 ($\pm$8.39) | 0.936 |
| FPG, mg dl$^{-1}$ | 92.61 ($\pm$10.39) | 90.95 ($\pm$9.10) | 0.378 |
| HOMA-IR | 4.14 ($\pm$3.56) | 3.58 ($\pm$1.75) | 0.492 |
| Insulin, mIU liter$^{-1}$ | 16.29 ($\pm$13.91) | 15.91 ($\pm$7.45) | 0.904 |
| TG, mg dl$^{-1}$ | 138.62 ($\pm$89.42) | 111.75 ($\pm$30.84) | 0.176 |
| T_Chol, mg dl$^{-1}$ | 189.59 ($\pm$40.46) | 185.62 ($\pm$29.53) | 0.699 |
| HDL-C, mg dl$^{-1}$ | 47.32 ($\pm$8.27) | 48.91 ($\pm$12.11) | 0.435 |
| LDL-C, mg dl$^{-1}$ | 116.96 ($\pm$35.61) | 115.78 ($\pm$25.55) | 0.878 |
| ALT, IU liter$^{-1}$ | 24.64 ($\pm$21.22) | 22.00 ($\pm$10.28) | 0.581 |
| AST, IU liter$^{-1}$ | 20.51 ($\pm$10.11) | 19.46 ($\pm$7.80) | 0.688 |
| Urea, mg dl$^{-1}$ | 13.46 ($\pm$2.05) | 13.23 ($\pm$3.09) | 0.757 |
| Cr, mg dl$^{-1}$ | 0.79 ($\pm$0.21) | 0.77 ($\pm$0.14) | 0.615 |
| UA, mg dl$^{-1}$ | 6.54 ($\pm$1.72) | 6.23 ($\pm$1.76) | 0.179 |

[a]BMI, body mass index; BMR, basal metabolic rate; WHR, waist-to-hip ratio; BFR, body fat ratio; VFA, visceral fat area; LBM, lean body mass; FPG, fasting plasma glucose; HOMA-IR, homeostasis model assessment-insulin resistance index; TG, triglyceride; T_Chol, total cholesterol; HDL-C, high-density lipoprotein cholesterol; LDL-C, low-density lipoprotein cholesterol; ALT, alanine aminotransferase; AST, aspartate aminotransferase; Cr, creatinine; UA, uric acid.
[b]Data are expressed as mean $\pm$ SD.
[c]$P$ values were determined by independent Student's $t$ test.
[d]$P$ value less than 0.05.

macronutrients and energy intake information provided by participants (24). The proportions of three macronutrients (e.g., fat, protein, and carbohydrate) in the ND and LCD groups of this study are summarized in Fig. 1B. The proportion of daily intake of carbohydrate was ~50% of the total in the ND group, while this ratio was reduced to ~20% for the LCD group over the whole study period (Fig. 1C). Moreover, energy calculated from daily dietary intake was generally higher in the ND group than in the LCD group (Fig. 1D). The carbohydrate proportion and daily energy intake in detail are summarized in Table S1A and S1B in the supplemental material. To assess the efficacy of weight loss for all participants, body composition and anthropometric parameters were analyzed. As expected, 12 weeks of LCD intervention significantly improved the parameters of body size (e.g., BMI, waist circumference, WHR, BFR, and VFA) (Table S1C). The reduction of BMI in the LCD group was up to 2.15 $\pm$ 1.24 kg m$^{-2}$ compared to that in the ND group, which was only 0.81 $\pm$ 0.69 kg m$^{-2}$ ($P <$ 0.001). In addition, a distinct decrease in waist circumference, WHR, BFR, and VFA in the LCD group was also observed (Fig. 1E). Parameters, such as glycometabolism, lipid metabolism, hepatic parameters, and renal function were not significantly different between the two groups (Table S1C and S1D).

**LCD intervention does not affect the overall microbial structure.** Other than distinct weight loss outcomes, various dietary components may affect the composition and diversity of gut microbiota, but other than overall composition and phylum-level changes, previous investigations did not reach a constructive conclusion to guide clinical trials of weight loss under LCD (25). In our study, fecal samples from all participants at baseline and end stage were collected for high-throughput sequencing. Through 16S rDNA gene-based analysis, 2,478,112 high-quality reads were obtained with an average of 23,828 reads (minimum, 12,583; maximum, 32,846; median, 23,833) per sample. The rarefaction measurement of the Shannon and Simpson indexes implied that sequencing depth captured all bacterial species and qualified for downstream analysis

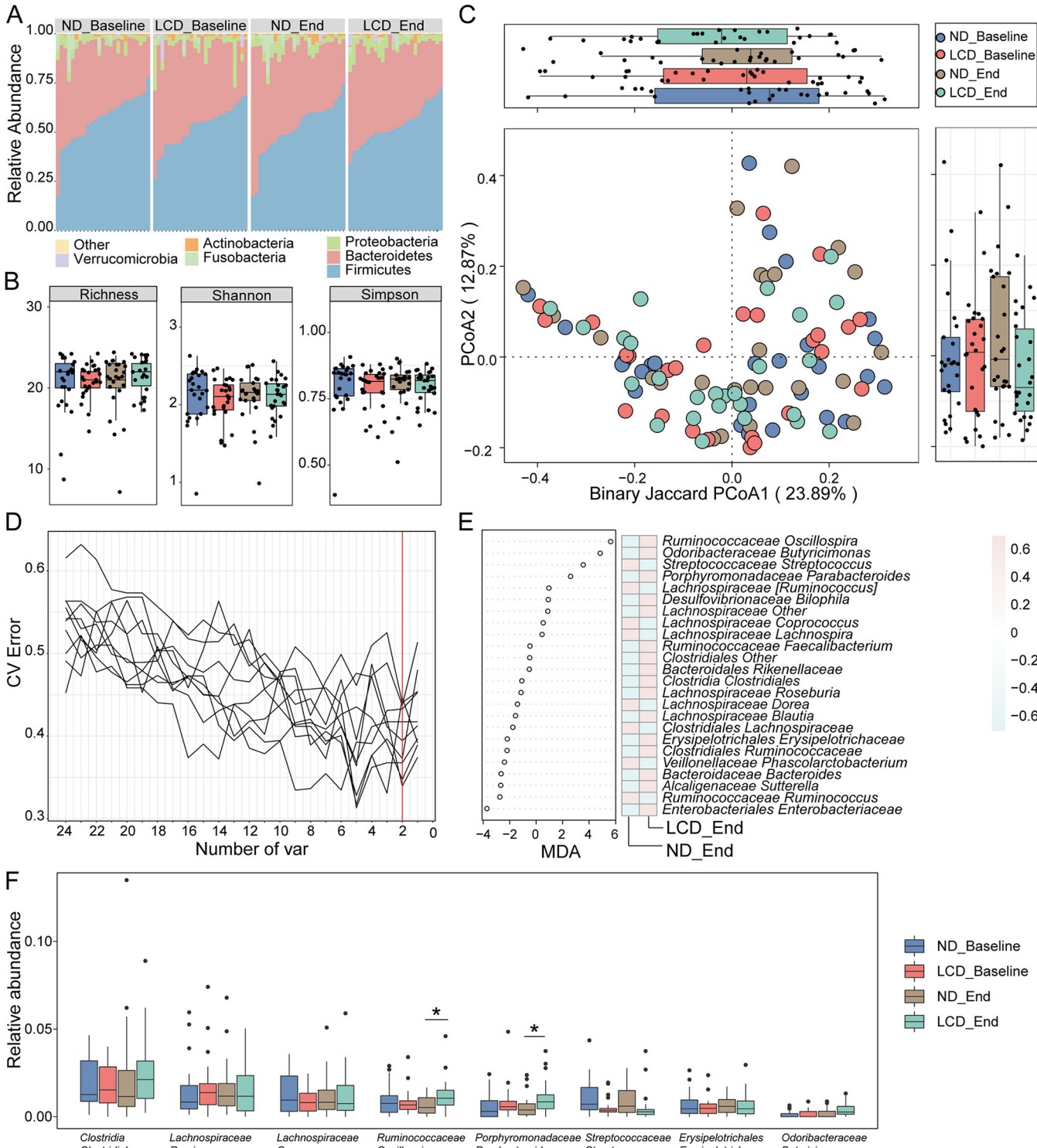

**FIG 2** Differential gut microbial characteristics in the ND and LCD groups at two different time points and potential bacterial markers of LCD intervention. (A) The overall composition and relative abundance of the bacterial community in each group at the phylum level were not significantly different. (B) Box plots of the $\alpha$-diversity index (richness, Shannon, and Simpson) showed no significant difference in $\alpha$-diversity indices between the ND group and LCD group at baseline stage or end stage. The horizontal lines in the box plots mean median values. The highest and the lowest boundaries of the box denote the 75% and 25% quartiles, and whiskers represent the lowest and highest values within 1.5 times the interquartile range (IQR) from the 25% and 75% quartiles, respectively. Dots represent data points beyond the whiskers. (C) The PCoA of $\beta$-diversity based on genus distribution by binary Jaccard algorithm showed that the gut taxonomic composition was not significantly different between the ND and LCD groups at the two different time points. (D) Two bacterial markers at the genus level were selected as optimal biomarkers of the random forest model in the ND and LCD groups after 12 weeks of dietary intervention. The red line illustrates the number of key bacteria in the discovery set. CV error, cross-validation error; var, variants. (E) The relative abundance of each bacteria at the genus level in the predictive model was assessed using

(Fig. S1A, B). The overall composition and abundance of the bacterial community at the phylum level in each group were not significantly different (Fig. 2A). To assess gut microbiota diversity, the $\alpha$-diversity values at the genus level, including the richness, Shannon, and Simpson indexes, were compared between two groups at the baseline and end stage (Fig. 2B). However, these indexes were not significantly different, suggesting that LCD intervention itself did not affect overall microbial composition and diversity. In addition, principal coordinate analysis (PCoA) based on binary Jaccard index at the genus level was performed to assess the $\beta$-diversity among different groups, and the microbiome structure between ND and LCD groups at two different time points was indistinct (Fig. 2C). These findings were consistent with previous studies; for example, a strict vegetarian diet intervention resulted in weight loss without a change in gut microbiota diversity (26). Other studies exhibited similar results of unaltered bacterial diversity, although they were not about dietary intervention or weight loss because there are many other confounding factors in the environment (27).

**LCD intervention efficacy is associated with a distinct group of bacterial biomarkers.** Here, we would like to raise the question of whether without an overall change in composition and diversity, a distinct group of bacteria is shifted with LCD intervention and contributes to the diet-host-microbiome interaction, which results in an obvious weight loss outcome. Therefore, to identify a particular group of bacteria, a predesigned machine learning algorithm was used. As reported before, we applied a 5-fold cross-validation together with a random forest algorithm to identify potential bacterial biomarkers with consideration of the lowest error rate plus standard deviation (28–30). With this hypothesis, 16S rDNA gene sequence data from ND and LCD groups before and after the clinical trial were further analyzed. For the data from the baseline stage of ND and LCD groups, 10 trials of analysis failed to identify biomarkers with significant differences in relative abundance between groups (Fig. S1C, D, S2A; Table S1E). After 12 weeks of LCD intervention, the same analysis was performed on the matched samples. We identified two potential bacterial biomarkers after 12 weeks of LCD intervention: *Ruminococcaceae Oscillospira* and *Odoribacteraceae Butyricimonas* (Fig. 2D and E). More specifically, the relative abundance of *Ruminococcaceae Oscillospira* was higher than that at baseline. Meanwhile, the relative abundance of *Odoribacteraceae Butyricimonas* had an increasing trend but did not reach statistical significance after 12 weeks of LCD intervention. Other than these, another bacterial biomarker was identified, *Porphyromonadaceae Parabacteroides*, that also had higher relative abundance after 12 weeks of LCD intervention (Fig. 2F; Fig. S2B). Meanwhile, the relative abundance of change of these key bacteria after LCD intervention positively correlates with clinical parameters, such as BMI, waist circumference, and BFR (Fig. S1E). Previous investigations have shown that these three bacteria are involved in butyrate production in the gut, indicating that an independent factor contributing to weight loss during LCD intervention may exist (31–33).

**Individual weight loss in each subgroup was different under two interventions.** Further analysis of weight loss outcome of changes in BMI, waist circumference, WHR, BFR, and VFA for each participant was performed through cluster stratification, and the median of these five anthropometric parameters was taken as the critical cutoff point. Two subgroups were defined: the moderate weight loss group (MG) and the distinct weight loss group (DG). For clinical characteristic analysis at baseline of both subgroups, only a slight difference in age for the LCD group was observed; the remaining characteristics did not show any significant difference (Fig. S3A, B and Table S1F, G). In terms of weight loss parameters for both subgroups (e.g., BMI, waist circumference, WHR, BFR,

**FIG 2** Legend (Continued)

mean decrease accuracy (MDA). The heat map illustrates the comparison of bacteria filtered by random forest via 5-fold cross-validation in the two groups at the end stage. (F) Box plots of all union optimal bacterial biomarkers selected through the random forest algorithm at baseline and end stages indicated that *Porphyromonadaceae Parabacteroides* and *Ruminococcaceae Oscillospira* were significantly increased in the LCD group after 12 weeks of LCD intervention. A two-way ANOVA with repeated measures followed by a Tukey *post hoc* test was used to compare multiple groups at different time points using GraphPad Prism 8.0.2; *, $P < 0.05$.

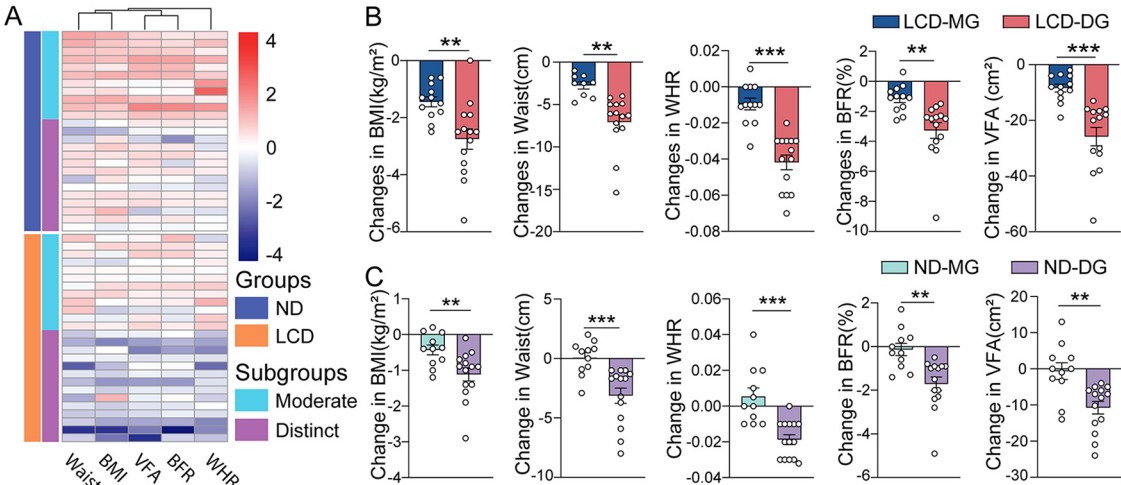

**FIG 3** Individual weight loss is varied under different dietary interventions. (A) Heat map clustered by changes in anthropometric parameters (waist, BMI, VFA, BFR, and WHR). Two groups were further classified into moderate weight loss groups (MG) and distinct weight loss groups (DG), respectively. (B) The obvious decrease in anthropometric parameters (BMI, waist, WHR, BFR, and VFA) of significant difference in LCD_DG compared to LCD_MG after 12 weeks of dietary intervention. Data are expressed as mean ± SEM and were analyzed by unpaired, two-sided Student's $t$ test; **, $P < 0.01$; ***, $P < 0.001$. (C) The slight reduction in anthropometric parameters (BMI, waist, WHR, BFR, and VFA) in ND_DG compared to that in ND_MG. Data are expressed as mean ± SEM and were analyzed by unpaired, two-sided Student's $t$ test; **, $P < 0.01$; ***, $P < 0.001$.

and VFA), both ND and LCD interventions showed significant changes (Fig. 3A). LCD presented a more dramatic reduction between MG and DG subgroups (BMI, 1.44 ± 0.62 versus 2.75 ± 1.34 kg m$^{-2}$; BFR [%], 3.28 ± 1.94 versus 7.05 ± 3.21; waist, 2.73 ± 1.11 versus 7.05 ± 3.21 cm; VFA, 8.39 ± 4.92 versus 25.83 ± 12.33 cm$^2$; WHR, 0.01 ± 0.01 versus 0.04 ± 0.02) (Fig. 3B; Table S1H, I). Moreover, ND intervention only showed a slight reduction between the two subgroups (BMI, 0.43 ± 0.46 versus 1.11 ± 0.7 kg m$^{-2}$; BFR [%], 0.14 ± 0.94 versus 1.71 ± 1.17; waist, 0.06 ± 1.43 versus 3.12 ± 2.42 cm; VFA, 0.64 ± 7.69 versus 10.73 ± 6.53 cm$^2$; WHR, −0.01 ± 0.02 versus 0.02 ± 0.01) (Fig. 3C; Table S1J, K). The energy intake and percentage of carbohydrates, fat, and protein in diets were almost the same between the two subgroups for ND and LCD intervention (Fig. S3C to F; Table S1L to Q). These data suggest that individual weight loss differences may be due to uncharacterized factors other than the percentage of carbohydrates in the diet.

**Microbial composition is a determining factor of distinct weight loss efficacy under LCD intervention.** LCD intervention results in effective weight loss efficacy, but variations can still be observed between individuals. Herein, we would like to ask whether there are any potential modulators leading to the difference in the two subgroups. It was shown that $\alpha$-diversity (e.g., richness, Shannon, or Simpson index) was not significantly different between the LCD_MG and LCD_DG subgroups at baseline or end stage (Fig. 4A). PCoA coupled with binary Jaccard showed that between the LCD_MG and LCD_DG subgroups, the microbial structure was significantly different at baseline (Fig. 4B) ($P = 0.0481$). Moreover, cooccurrence analysis was performed to explore the interaction between gut microbiota in LCD subgroups. Although the network interaction complexity was decreased in both LCD_MG and LCD_DG subgroups with LCD intervention for 12 weeks, the LCD_DG group exhibited stronger and broader network interaction complexity than the LCD_MG group (Fig. 4C to F). The alteration in subcommunity networks suggested that other than the difference in composition and diversity, microbial differences in structure and complexity could partially explain the inconsistent outcomes of LCD.

**Identification of microbial biomarkers that interact with LCD intervention.** To date, optimal clinically assessable biomarkers to guide weight loss under LCD intervention have not yet been defined. To disclose the mystery, we applied 5-fold cross-validation together with random forest to 16S rDNA gene sequence data at the baseline and

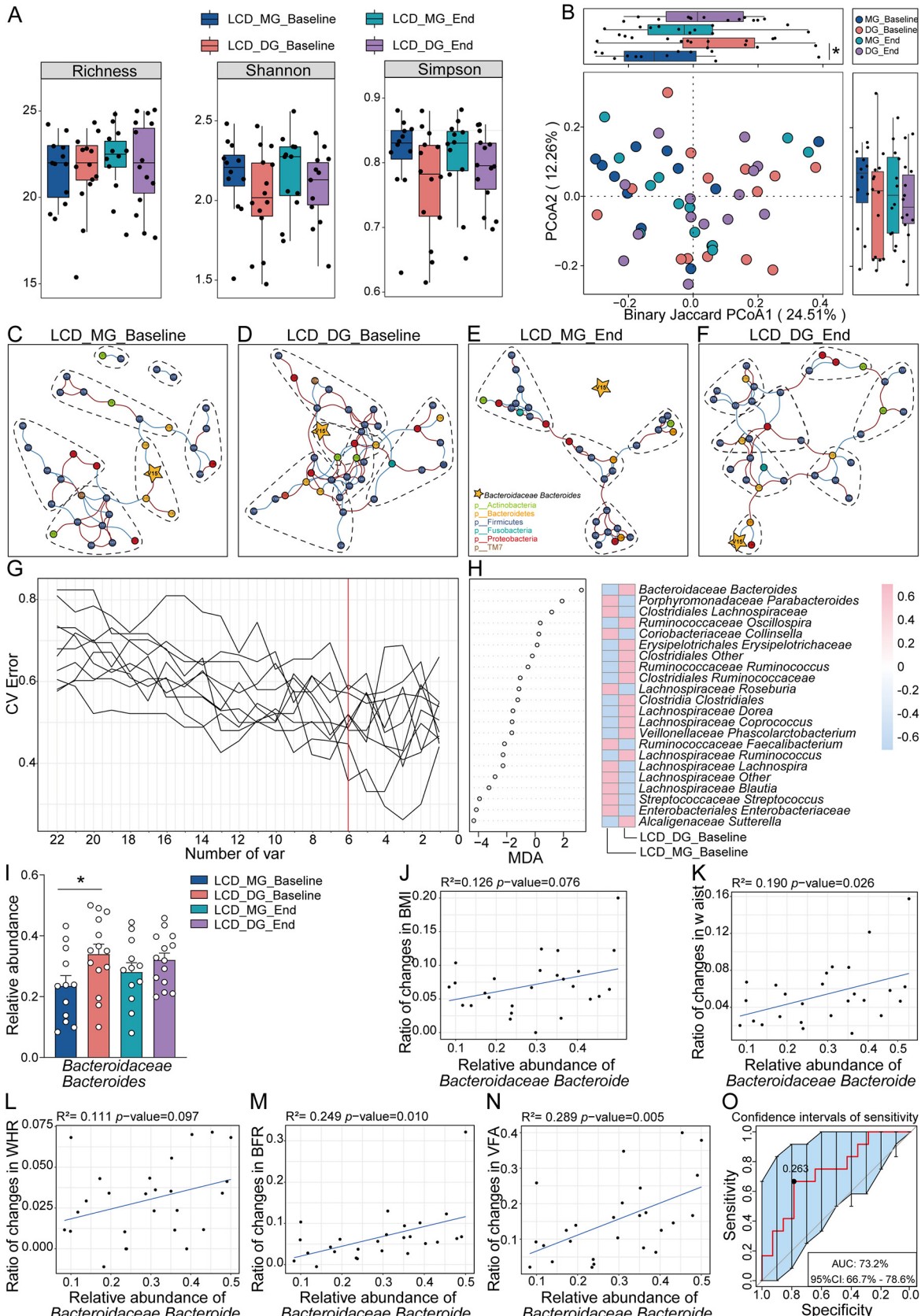

**FIG 4** Microbial composition is a determining factor regarding distinct weight loss efficacy under LCD intervention. (A) Box plots of the α-diversity index (richness, Shannon, and Simpson) at the genus level showed no significant difference between LCD subgroups at

end stages. Six optimal biomarkers were identified between the LCD_MG and LCD_DG subgroups at baseline (Fig. 4G and H; Fig. S4A). However, only one bacterial biomarker showed a significant difference in relative abundance at baseline ($P$ = 0.037) (Fig. 4I). The relative abundance of *Bacteroidaceae Bacteroides* at baseline positively correlated with the ratio change of weight loss parameters at the end stage (BMI, $R^2$ = 0.126, $P$ = 0.076; waist, $R^2$ = 0.190, $P$ = 0.026; WHR, $R^2$ =0.111, $P$ = 0.026; BFR, $R^2$ = 0.249, $P$ = 0.010; VFA, $R^2$ = 0.289, $P$ = 0.005) (Fig. 4J to N). Thus, the predictive linear regression model based on the relative abundance of *Bacteroidaceae Bacteroides* could achieve an area under the curve (AUC) value of 73.2% with a confidence interval (CI) of 66.7 to 78.6% between LCD_MG and LCD_DG subgroups to predict the outcome of weight loss effi-cacy (Fig. 4O). The difference did not remain after LCD intervention between the MG and DG subgroups (Fig. S4B and S5A to C).

**The ANN model to predict the outcome of weight loss efficacy in the LCD group.** Because of the intricate connections between bacteria, the predicted perform-ance only focusing on *Bacteroidaceae Bacteroides* was not sufficient for prediction in a clinical setting. To overcome this shortcoming, we integrated an artificial neural net-work (ANN) model, an even more robust deep learning model that is trained and used to imitate biological neural networks. In recent years, an increasing number of medical studies have applied the ANN model to process complex data because of its superior-ity, such as to seek predictors of catheter-related thrombosis in hospitalized infants (34), to calibrate the prediction of survival in glioblastoma patients (35), and to opti-mize the auxiliary diagnosis of insomnia disorder (30). By integrating the data of the LCD group into our ANN model, including clinical anthropometric parameters and fil-tered relative abundance of all microbiota at the genus level, this model could result in a high coefficient of determination. Based on the anthropometric parameters at base-line (BMI, waist circumference, WHR, BFR, and VFA), change of anthropometric parame-ters or the ratio of change over baseline parameters, our ANN model reached a high prediction rate (Fig. 5A to O) (changes in BMI, $R^2$ = 0.307, mean absolute error [MAE] = 0.780; changes in waist circumference, $R^2$ = 0.316, MAE = 1.869; changes in WHR, $R^2$ = 0.491, MAE = 0.010; changes in BFR, $R^2$ = 0.470, MAE = 0.980; changes in VFA, $R^2$ = 0.322, MAE = 7.705; ratio of changes in BMI, $R^2$ = 0.344, MAE = 0.022; ratio of changes in waist circumference, $R^2$ = 0.219, MAE = 0.020; ratio of changes in WHR, $R^2$ = 0.449, MAE = 0.013; ratio of changes in BFR, $R^2$ = 0.577, MAE = 0.028; ratio of changes in VFA, $R^2$ = 0.571, MAE = 0.051).

## DISCUSSION

Our study confirmed that overweight or obese people achieved significant weight loss on a short-term LCD intervention of *ad libitum* energy intake without causing clear adverse effects, which was consistent with previous studies (12, 36). After 12 weeks of LCD intervention, the relative abundances of certain butyrate-producing bacteria were dramatically elevated. Indeed, the correlation between LCD intervention and gut

**FIG 4** Legend (Continued)

baseline stage or end stage. (B) The PCoA of $\beta$-diversity based on genus distribution by binary Jaccard algorithm showed that the gut taxonomic composition was significantly different between LCD subgroups at baseline but not end stage; *, $P$ = 0.0481 is from least significant difference (LSD). (C to F) The cooccurrence networks before and after LCD intervention reflect network interaction complexity. All nodes were colored at the phylum level (isolated nodes were excluded), and edges were estimated by Spearman's rank correlation coefficient (abs[r] > 0.3, $P$ < 0.05). On the whole, LCD_DG exhibited stronger and broader network interaction complexity than LCD_MG at two different time points. (G) Six markers at the genus level were selected as optimal biomarkers of the random forest model in LCD subgroups at baseline. The red line illustrates the number of key bacteria in the discovery set. (H) The relative abundance of each bacteria at the genus level in the predictive model was assessed by MDA. The heat map illustrates the comparison of bacteria filtered by random forest via 5-fold cross-validation in the two subgroups at the baseline stage. (I) The relative abundance of *Bacteroidaceae Bacteroides*, selected through the random forest, was significantly higher in LCD_DG than in LCD_MD at the baseline stage. Data are expressed as mean ± SEM, and a two-way ANOVA with repeated measures followed by a Tukey *post hoc* test was used to compare multiple groups at different time points using GraphPad Prism 8.0.2; *, $P$ < 0.05. (J to N) Linear regression indicates that the relative abundance of *Bacteroidaceae Bacteroides* was positively correlated with the ratio of changes in weight loss parameters (BMI, waist, WHR, BFR, and VFA). (O) The baseline relative abundance of *Bacteroidaceae Bacteroides* achieved an AUC value of 73.2% with a 95% confidence interval (95% CI) of 66.7% to 78.6% between LCD_MG and LCD_DG to predict the outcome of weight loss efficacy.

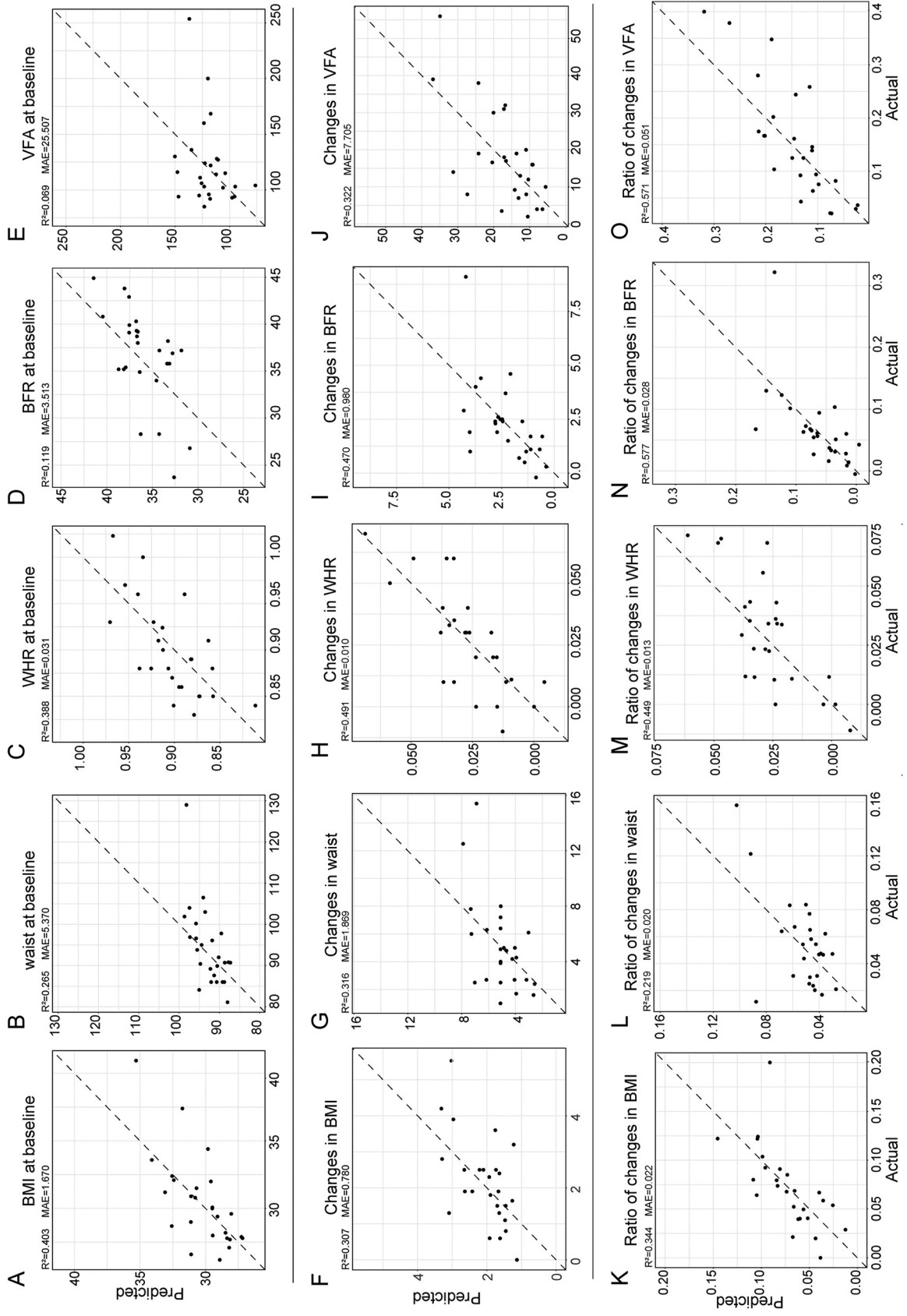

**FIG 5** Gut microbiota-based prediction of the clinical weight loss parameters after LCD treatment by ANN. A high-accuracy ANN prediction model was established using the relative abundance of gut microbiota at the genus level and weight loss parameters, including the baseline value of BMI, waist, WHR, BFR, and VFA after LCD treatment (A to E), changes in weight loss parameters (F to J), and the ratio of changes in these parameters to their baseline value in 5-fold cross-validation (K to O).

microbiota could be bilateral, as previously demonstrated by others, suggesting that gut microbiota may have contributed to weight loss outcome under LCD intervention. Further analysis demonstrated that a higher relative abundance of *Bacteroidaceae Bacteroides* at baseline was positively correlated with weight loss efficacy. Most importantly, our ANN prediction model with high accuracy has proven that the microbiota at baseline can act as a predictor to assess individualized weight loss outcomes before LCD intervention.

LCD has a long history since being first acknowledged in the 1860s, but the contribution of LCD to obesity remains elusive (37). The literature has no clear consensus or definition about what amount of carbohydrates should be consumed every day (37). There is little evidence for the superiority of greater carbohydrate restriction. Additionally, excessive restriction of carbohydrates, such as only 5% of total energy driven from carbohydrates, may not be realistically achievable for most participants (23, 38). Harvey and coworkers (38) indicated that an LCD containing 15% of total energy from carbohydrates was easily adhered to over a 12-week period and can achieve acceptable improvements in weight loss. However, long-term adherence to an LCD should be advocated with great caution to avoid an increased risk of colonic disease (25) and a potential impact on coronary atherosclerosis (39). Regarding compliance, safety, and cost, we designed this 12-week LCD intervention study in line with Harvey's research (38). The application of nutrition bars and guidance from nutritionists allowed us to modulate the consumption of carbohydrate scientifically and effectively for every participant in this study. One article reported that the cardiorespiratory fitness and cardiometabolic profiles of obese individuals could be improved through short-term LCD combined with prescribed exercise (40). As we know, physical exercise is an efficient combined strategy for bodyweight control. Therefore, we recruited participants who engaged in light physical activity or work and kept exercise time to approximately 60 min per week as much as possible to minimize the influence of confounding factors. Although energy intake was not restricted, the results of the study showed that the LCD group generally had lower energy intake than the ND group. Such an observation might be explained by the higher satiety of protein as well as hormonal regulation effects of an LCD. Since the carbohydrate content of the diets is significantly reduced, the relative proportion of energy derived from protein and fat increased among these three main nutrients (41). It was reported that higher protein intake due to an LCD may also increase satiety, resulting in decreased overall energy intake. In addition, an LCD may influence hormones that could impact hunger and appetite control, such as ghrelin, leptin, and cholecystokinin, although researchers did not reach a consensus (23). Collectively, these effects may help to explain the lower energy intake in the LCD group in our study (12). However, an LCD causes preferential body fat loss in comparison with isocaloric, higher-carbohydrate diets, which might be due to increased adipose lipolysis and fat oxidation as well as less fat synthesis (12). Moreover, Bravata et al. reported that diets rich in protein and short of carbohydrates could achieve rapid weight loss without significant adverse effects by promoting the metabolism of adipose tissue in the absence of available dietary carbohydrates (37). The present study showed that a short-term LCD intervention results in significant weight loss without causing adverse effects on liver and kidney functions or glycolipid metabolism in the participants, which was in accordance with other reports. For the management of other influencing factors, such as probiotics, numerous studies have reported that probiotic supplementation improves obesity-related parameters, leading to weight loss (42). Herein, in our clinical trial, no probiotics or prebiotics were allowed during the experimental period.

Excluding the effect of taking probiotics and prebiotics, we surprisingly found that short-term LCD intervention increased the relative abundance of certain groups of gut bacteria that were positively associated with weight loss. A weight loss diet can alter the composition of the human gut microbiota, which is highly variable (43). An LCD has long been controversial regarding the impact on gut microbiota. Russell et al. reported that

*Bacteroidetes* were decreased in people with obesity on a high-protein, low-carbohydrate diet for 4 weeks (25). In another study, the authors found significant diet-dependent reductions in a group of butyrate-producing *Firmicutes* in fecal samples from obese participants on a low-carbohydrate, weight-reducing diet for 4 weeks, but no changes were found in the abundance of *Bacteroidetes* (43). After 12 weeks of an LCD intervention, the gut microbiota was highly enriched for the genera *Porphyromonadaceae Parabacteroides* belonging to *Bacteroidetes* and *Ruminococcaceae Oscillospira* belonging to *Firmicutes*. Previous studies have proven that these three bacteria are involved in producing butyrate in the gut (32, 33, 44). However, the role of butyrate in glycolipid metabolism is still controversial. Duncan and coworkers found that a reduction in dietary carbohydrate intake caused a decrease in the concentration of butyrate and butyrate-producing bacteria in the feces of participants with obesity (45). In contrast, some other studies reported that butyrate stimulates gut hormones (e.g., glucagon-like peptide-1 [GLP-1]) and restrains food intake to alleviate obesity (46).

Among the above dominant genera, bacteria of the *Parabacteroides* genus are saccharolytic and produce the major end products of fermentation, such as acetic acid and succinic acid (47). The relative abundance of *Parabacteroides* was significantly negatively correlated with BMI. Interestingly, members of the *Parabacteroides* family, such as *Parabacteroides goldsteinii* (48) and *Parabacteroides distasonis* (49), are promising probiotics that could alleviate obesity and obesity-associated metabolic dysfunctions. However, it was reported that *Oscillospira* was associated with leanness or lower BMI and was significantly more abundant in metabolically healthy participants who were overweight or obese (50). One study showed that members of the *Oscillospira* genus are highly heritable and positively associated with the leanness-promoting bacterial species *Christensenella minuta*. Animal experiments confirmed that GF mice gain less weight after receiving obese donor microbiota spiked with *C. minuta* along with enrichment of *Oscillospira* (51). Some *Oscillospira* species likely could secrete important short-chain fatty acids (SCFAs), which are beneficial for body weight control as well as glucose and lipid homeostasis (44). Another possible reason for the association between *Oscillospira* and leanness was that *Oscillospira* may be able to degrade host glycans and thus help hosts spend metabolic energy to regenerate degraded glycoproteins (50). In this regard, we speculated that the increased abundance of *Porphyromonadaceae Parabacteroides* and *Ruminococcaceae Oscillospira* in this study may be a response of gut microbiota to dietary intervention, assisting in weight loss in the LCD.

Interpersonal differences in weight loss within the LCD group have further emphasized the role of gut microbiota in LCD interventions. Participants in the LCD_DG subgroup achieved significant weight loss compared to participants in the LCD_MG subgroup, although both of them had similar proportions of carbohydrates, fat, and protein in LCD patterns. Moreover, our work demonstrated that a higher relative abundance of *Bacteroidaceae Bacteroides* at baseline was significantly associated with superior weight loss after a short-term LCD intervention. The gut microbiota is a complex and dynamic ecosystem changing with modifiable aspects (52). In the present study, the microbiota in participants who achieved more distinct weight loss exhibited stronger and broader network interaction complexity than the microbiota of others in both LCD groups. In particular, at the end stage of the trial, as the complexity of the network decreased, *Bacteroidaceae Bacteroides* (labeled with a star in Fig. 5C to E) was isolated from the network community in the LCD_MG subgroup. To determine whether *Bacteroidaceae Bacteroides* could be discriminated out of weight loss on an LCD intervention, a correlation analysis and a receiver operating characteristic (ROC) curve analysis were performed. From the results of these analyses, we found that the relative abundance of *Bacteroidaceae Bacteroides* at baseline was positively correlated with the ratio change in weight loss parameters at the end stage. *Bacteroides* is one genus of the dominate microbiota that comprises the majority of the bacterial taxa in the gut of most individuals (53). A previous study confirmed that individualized gut mucosal colonization capacity correlated with baseline host transcriptional and microbiome characteristics. In addition, probiotic colonization is predictable by the pretreatment

microbiome. In summary, the baseline gut microbiota composition plays an essential role in the metabolism of the host (54). Moreover, our result is in line with a recent study that reported that the baseline gut microbiota was a preeminent predictor of individual weight loss trajectories. More precisely, the author proved that *Bacteroides dorei* was one of the strongest predictors for weight loss when present in high abundance at baseline (55). Similarly, another study reported that the baseline relative abundance of a common species in the human gut, *Bacteroides cellulosilyticus*, was the most important predictor of body weight gain among the top 10 *Bacteroidetes* species during the intervention of arabinoxylan-oligosaccharides (AXOS) (10.4 g day$^{-1}$) from wheat bran or polyunsaturated fatty acids (PUFAs) (3.6 g day$^{-1}$) (56). Therefore, some members belonging to *Bacteroidaceae Bacteroides* are of great importance in host metabolism and promote weight loss following dietary intervention.

However, the regression $R^2$ value of the correlation analysis in the present study was unsatisfactory compared to our anticipated goal. In view of the complexity of the gut microbiota ecosystem, we applied an even more powerful deep learning model, called ANN, to improve the accuracy of the prediction model. ANNs are trained and used to imitate biological neural networks since they include a set of computational nodes and generate signals transmitted from neuron to neuron (30, 34). In recent years, an increasing number of medical studies have applied the ANN model to process complex data because of its superiority, such as to seek predictors of catheter-related thrombosis in hospitalized infants (34) and to optimize the auxiliary diagnosis of insomnia disorder (30). In the present study, by integrating the data of LCD subgroups into the ANN model, including clinical anthropometric parameters and filtered relative abundance of all microbiota at the genus level, the results of ANN exhibited a larger $R^2$ value of each parameter, signifying higher accuracy for prediction. Similar to a previous study, the predictive function of discriminatory species was improved with the interaction of multiple factors (57). We demonstrated that *Bacteroidaceae Bacteroides* may play causal roles in the prediction of weight loss in an LCD intervention while other bacteria play auxiliary roles. From what has been discussed above, the relative abundance of gut microbiota at baseline can be a powerful predictor of weight loss outcome after LCD intervention. An individualized weight loss prediction model based on the baseline relative abundance of gut microbiota may be applied to clinical practice in the future.

In summary, short-term LCD intervention without calorie restriction can produce significant weight loss in overweight and obese populations. Differences in the gut microbiota contributed to inconsistent weight loss outcome on LCD intervention. The relative abundance of gut microbiota at baseline may be taken into account in clinical practice when evaluating the applicability of LCD intervention for weight loss. However, it may be worth exploring probiotics for *Bacteroidaceae Bacteroide* that can be individually added when using LCD interventions for weight loss. Nevertheless, this might provide new hints in drug discovery and change the landscape of diet intervention in the near future.

## CONCLUSION

Short-term LCD intervention of *ad libitum* energy intake facilitates weight loss. The relative abundances of certain butyrate-producing bacteria were dramatically elevated after 12 weeks of LCD intervention, indicating that gut microbiota contributed to weight loss outcome under LCD intervention. A higher relative abundance of *Bacteroidaceae Bacteroides* at baseline was positively correlated with weight loss efficacy. Most importantly, our ANN prediction model with high accuracy based on the relative abundance of all bacteria at baseline proved that the microbiota at baseline can act as predictors to assess individualized weight loss outcomes before LCD intervention. In future studies, it will be worthwhile to test our prediction model in a large cohort to further prove that gut microbiota at the baseline could be utilized to predict the weight loss outcome after LCD intervention.

The present investigation has several limitations. First, the average body weight of participants in the LCD group was slightly heavier (BMI for ND and LCD, 28.61 ± 2.04 versus 30.44 ± 3.38 kg m$^{-2}$) than that in the ND group at the baseline, which was indeed a disadvantage of this study, but it is not easy to avoid. As all participants were

randomly enrolled in this study, it was difficult to ensure that the participants' weight was the exact same between two groups at baseline. Second, we found that the relative abundance of *Ruminococcaceae Oscillospira* and *Odoribacteraceae Butyricimonas* as well as *Porphyromonadaceae Parabacteroides* increased after LCD intervention for 12 weeks. According to the literature review, these three kinds of microbiota were related to the production of butyrate in the gastrointestinal tract. In addition, previous studies show that *Bacteroidetes* are the largest propionate producers in the human gut. However, we failed to conduct fecal metabonomic detection due to the exhausted samples, so the content of butyrate or propionate in the intestine of the participants could not be determined so as to confirm the 16S rDNA gene sequencing result of the relative abundance of *Bacteroides* by quantitative PCR (qPCR).

## MATERIALS AND METHODS

**Participants.** This clinical trial was approved by the Institutional Review Board in Zhujiang Hospital of Southern Medical University. The study was conducted according to the Declaration of Helsinki guidelines and was registered in the China Clinical Trial Registry (clinical trial approval no. ChiCTR1800015156). For the present study, 51 overweight or obese individuals (BMI between 26.2 and 40.94 kg m$^{-2}$, age from 21 to 59 years old) were recruited. The diagnosis for being overweight and obese was based on the diagnostic criteria present in reference 58. In short, being overweight was defined as 24 kg m$^{-2}$ $\leq$ BMI<28 kg m$^{-2}$, while a BMI of $\geq$28 kg m$^{-2}$ was classified as obese.

Inclusion criteria were (i) body weight of the participants was changed steadily or without significant change in the last 30 days, (ii) no antibiotics or drugs were taken either 3 months before or during the course of the weight loss trial, and (iii) all participants voluntarily participate in and cooperate with the research investigator and sign the informed consent form before the formal start of the trial. Exclusion criteria were (i) pregnancy or preparing for pregnancy during the study period, (ii) weight loss treatment application in the past 30 days, (iii) history of gastrointestinal disease or surgery, (iv) type 1 diabetes, type 2 diabetes, Cushing or Cushing syndrome, thyroid-related diseases and other endocrine system diseases, (v) severe hypertension or any other cardiovascular diseases in the past 6 months, (vi) clinical hepatobiliary diseases, including but not limited to chronic active hepatitis and/or severe liver insufficiency and cirrhosis, (vii) kidney disease history, (viii) history of acute or chronic infection, surgery, or severe trauma in the past 6 months, (ix) adopting any other experimental drugs within the past 30 days, (x) alcohol or drug abuse in the past 6 months, (xi) heavy manual labor, (xii) serious physical or psychological diseases, (xiii) malignant tumors, multiple organ dysfunction, immunocompromised function, and (xiv) participants who cannot comply with the protocol.

**General study design.** The eligible participants were screened according to the inclusion and exclusion criteria during the screening period 2 weeks before the baseline. Dietary overview handouts, sample menus, recommended recipes, and a book to calculate calories and carbohydrates were distributed to the participants 10 days before the baseline stage guided by a nutrition consultant. The clinical data of the participants were reevaluated by investigators to confirm whether the participants were qualified before the baseline stage. The study lasted for 12 weeks, and the baseline was the first day of the trial. The grouping and dietary intervention plans were determined according to participants' unique random number. During the trial period, participants attended eight face-to-face visits at the study center (1 visit per week for the first 4 weeks and then visits in the 6th, 8th, 10th, and 12th weeks subsequently). At other times, participants were contacted by nutritionists or investigators by phone to supervise their diets, to ensure compliance, and to record observed side effects in time. Samples of feces and blood were collected at the baseline stage and end stage for later assays of fecal 16S rDNA gene sequencing as well as hematic biochemical index detection. After that, further analysis of clinical data and 16S rDNA gene sequencing data analyses were performed.

**Dietary intervention.** Participants in this experiment were grouped randomly with a random assignment code that was generated by a random number generator. The serial number corresponds to the participants' random assignment code. A sealed letter was printed for each serial number and sent to each test center. After the researchers screened the participants who met the conditions of this study, they opened the corresponding sealed letters and selected the methods corresponding to the serial numbers for the intervention. Participants were randomly assigned to either the ND or LCD group both with *ad libitum* energy intake.

For the ND group (the normal diet without energy restriction group), according to the dietary guidelines of Chinese residents, the staple food should be at least 240 g (800 kcal), and the calories provided by carbohydrates should account for 55 to 65% of the total calories.

In the LCD group, participants had fixed low-carbohydrate diets (10 to 25% of total energy intake) (23, 38) without calorie restriction. Participants adopted standardized nutrient bar replacement to ensure the LCD structure. Additionally, vegetable oil was chosen as the main cooking oil. Taboo foods included fruits, bread, pasta, and other grains, processed meats, such as bacon and lunch meat, and high-fat red meat, such as pork, fatty beef, fatty lamb, and poultry. Participants were allowed to freely eat except the taboo foods for breakfast. For lunch and dinner, standardized nutrient bar replacements were provided, respectively. The components of the nutrition food bar are presented in Table S1R (see supplemental materials). In addition, participants were free to eat food other than taboo foods at lunch and dinner. Participants in this group accepted the LCD intervention for 12 weeks.

Every participant's diet was supervised by a dietitian to determine whether it was standard. The 3-day 24-h dietary recalls provided by participants every week were used to calculate the average energy

intake and proportion of carbohydrates. The energy consumed by participants was calculated according to Chinese food composition tables (24).

**Sample and clinical data collection.** The age, sex, and vital signs of the participants were recorded in detail. Weight, height, waist circumference, and hip circumference were measured on an empty stomach in the morning without shoes and wearing a single layer of clothes. Anthropometric parameters, such as BMI, BFR, WHR, VFA, and basal metabolic rate (BMR) were measured by an IOI353 body composition analyzer (Jawon Medica Ltd., South Korea.). Blood samples were collected both at the baseline and end stage. Biochemical indicators of the blood were completed by the laboratory. Detail indexes were as follows: glycometabolism index for fasting plasma glucose (FPG), fasting insulin; lipid metabolism index such as high-density lipoprotein cholesterol (HDL-C), low-density lipoprotein cholesterol (LDL-C), triglyceride (TG), and total cholesterol (T_Chol); hepatic function index (e.g., alanine aminotransferase [ALT] and aspartate aminotransferase [AST]); and renal function index (e.g., urea, creatinine [Cr], and uric acid [UA]). Some parameters were obtained by calculation, for example, insulin resistance index (HOMA-IR) = fasting blood glucose ([mmol liter$^{-1}$] $\times$ fasting insulin [mIU liter$^{-1}$]/22.5). Laboratory inspection items were completed on the morning of the visit after an overnight fast for at least 10 h, and no vigorous exercise was allowed the day before. Moreover, participants were guided to collect fecal samples by themselves before the formal start of the trial. Therefore, fecal samples were autonomously collected by participants at baseline and end stage. Fecal samples were stored in a $-80°$C freezer.

**Fecal DNA extraction and sequencing.** Microbial DNA extraction of fecal samples was performed according to the instructions of the ZR Fecal DNA kit (Zymo Research, USA). All samples were sequenced on an Illumina HiSeq 2500 platform.

**Bioinformatics and statistics.** The sequences of the 16S rDNA gene V3-V4 region were amplified on an Illumina high-throughput sequencing platform. Chimera-free sequences filtered by VSEARCH were applied to a standard QIIME 1.91 pipeline (59). A total of 2,478,112 high-quality reads with an average of 23,828 reads (minimum, 12,583; maximum, 32,846; median, 23,833) per sample were obtained. Operational taxonomic units (OTUs) at a 97% similarity threshold were clustered by the "Open-Reference" approach, and taxonomy profiles were mapped using the RDP classifier against Greengenes version 13.5 database (60). For all subsequent analyses, interfering taxa were applied to discard those whose relative abundance presented below 0.1% in at least 70% of participants in each group. Regular alpha rarefaction, including Shannon, Simpson, and richness indexes, was calculated with the R package 'VEGAN'. $\beta$-Diversity was computed on Bray-Curtis distance and estimated in two-dimensional space in the R packages 'VEGAN' and 'ggplot2'. The cooccurrence analysis based on the 'igraph' package was calculated for the bacterial network at the genus level and determined subgroups by the fast-greedy modularity optimization algorithm. All nodes were colored at the phylum level (isolated nodes were excluded), and edges were estimated by Spearman's rank correlation coefficient (abs[r] $>$ 0.3, $P <$ 0.05). To estimate the importance scores of each taxon, 5-fold cross-validation together with random forest analysis were incorporated to probe key signature microbiota by the R package 'randomForest'. An ROC curve was plotted with the 'pROC' package. This study used innovative strategies, including grid search and 5-fold cross-validation, to train an ANN prediction model utilizing pyTorch, sklearn, and numpy packages and selected the optimized parameters consisting of learning rate, activation function, layers, neurons, and dropout.

**Other statistical analyses.** To determine differences in clinical data between study groups, a statistical analysis was performed using IBM SPSS Statistics 20. The results are expressed as the means $\pm$ standard deviation (SD) in the table or the means $\pm$ standard error of the mean (SEM) for bar graphs. Independent samples $t$ tests were used to compare the two groups at baseline. Changes within group of clinical characteristics at the end stage were analyzed by using paired samples Student's $t$ tests. The two-way analysis of variance (ANOVA) with repeated measures followed by a Tukey *post hoc* test was performed to compare the continuous variables, including anthropometric parameters (BMI, waist circumference, WHR, BFR, and VFA), in multiple groups by GraphPad Prism 8.0.2. $P$ values less than or equal to 0.05 were considered significant in the study.

**Data availability.** Supporting information is available from the author. 16S rDNA gene sequencing data were deposited at NCBI with project number PRJNA752174.

## SUPPLEMENTAL MATERIAL

Supplemental material is available online only.

**SUPPLEMENTAL FILE 1**, XLSX file, 0.1 MB.
**SUPPLEMENTAL FILE 2**, PDF file, 2.3 MB.

## ACKNOWLEDGMENTS

We thank all participants for their active cooperation. The authors also would like to thank Guangzhou Nanda Fit Nutrition and Health Consulting Co., Ltd., for donations of nutrition food bars.

This work was supported by research grants from the Natural Science Foundation of China (grant no. 81770835, 81974117, 81900797, 82072436, and 81774035), the Guangdong Basic and Applied Basic Research Foundation (grant no. 2019A1515010665 and 2020B1515020046), and 'GDAS' Project of Science and Technology Development (grant no. 2018GDASCX-0102 and 2021GDASYL-20210102003).

Liwei Xie and Susu Zhang are also the inventors for the patent 2021104655623.
We declare no conflicts of interest.

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
