## [Reviewer comments · Microbiology Spectrum]

Microbiology Spectrum

Gut microbiota serves a predictable outcome of short-term low-carbohydrate diet (LCD) intervention for patients with obesity

Susu Zhang, Peili Wu, Ye Tian, Bingdong Liu, Liuqing Huang, Zhihong Liu, Nie Lin, Ningning Xu, Yuting Ruan, Zhen Zhang, Ming Wang, Zongbin Cui, Hong-Wei Zhou, Liwei Xie, Hong Chen, and Jia Sun

Corresponding Author(s): Liwei Xie, Guangdong Provincial Key Laboratory of Microbial Culture Collection and Application, State Key Laboratory of Applied Microbiology Southern China, Institute of Microbiology, Guangdong Academy of Sciences, Guangzhou, 510070, China

Review Timeline:

Submission Date:	May 7, 2021
Editorial Decision:	June 24, 2021
Revision Received:	July 22, 2021
Editorial Decision:	August 4, 2021
Revision Received:	August 9, 2021
Accepted:	August 15, 2021

Editor: Steven Frese

Reviewer(s): Disclosure of reviewer identity is with reference to reviewer comments included in decision letter(s). The following individuals involved in review of your submission have agreed to reveal their identity: Mohammed Hankir (Reviewer #1)

Transaction Report:

DOI: <https://doi.org/10.1128/Spectrum.00223-21>

June 24, 2021

Dr. Liwei Xie
Guangdong Institute of Microbiology
Guangdong, Guangzhou 510070
China

Re: Spectrum00223-21 (Gut microbiota serves a predictable outcome of short-term low-carbohydrate diet (LCD) intervention for patients with obesity)

Dear Dr. Liwei Xie:

Please

Thank you for submitting your manuscript to Microbiology Spectrum. When submitting the revised version of your paper, please provide (1) point-by-point responses to the issues raised by the reviewers as file type "Response to Reviewers," not in your cover letter, and (2) a PDF file that indicates the changes from the original submission (by highlighting or underlining the changes) as file type "Marked Up Manuscript - For Review Only". Please use this link to submit your revised manuscript - we strongly recommend that you submit your paper within the next 60 days or reach out to me. Detailed information on submitting your revised paper are below.

To comply with the Journal's Data Availability policy and to receive acceptance of the paper, please add a Data Availability statement and accession numbers for sequencing data generated in this study to the revised manuscript.

Link Not Available

Sincerely,

Steven Frese

Journals Department
Reviewer comments:

Reviewer #1 (Comments for the Author):

Zhang et al aim to identify if differences in gut microbiota composition account for the variable outcomes of LCD in a group of 51 overweight/obese Chinese patients. They find that LCD (10-25% calories from carbohydrates according to NLA guidelines) for 12 weeks resulted in greater changes in BMI and other adiposity indices than normal diet associated with reduced overall energy intake. This was independent of any major changes in fecal microbiota. The authors then show using a machine learning algorithm that 3 distinct gut microbiota species that produce butyrate are increased in the LCD group after intervention. Additionally, they find that Bacteroidaceae Bacteroides abundance at baseline is higher in the distinct weight loss LCD subgroup compared with the moderate weight loss subgroup and that this had greater predictive power using ANN analysis.

This is an excellent study that is very well presented and executed and provides significant new insight into why the efficacy of LCD varies between people. My only real concern is the difference in BW at baseline between groups which needs to be mentioned in the discussion as a limitation. There is also a little ambiguity about which class of bacteria could be contributing more to the outcomes of LCD. Is it the increase in SCFA-producing bacteria during intervention or is it the high levels of Bacteroidaceae Bacteroides at baseline or is it both? Maybe a schematic diagram could help clarify this issue. Further from this point, while the role of SCFA including butyrate in regulating energy balance is discussed, what factors produced by Bacteroidaceae Bacteroides could contribute to the outcome of LCD? Finally, if the authors cannot measure SCFA in fecal samples, this needs to be mentioned as another limitation. I have the following minor comments/suggestions: Introduction, line 70: As causes of obesity, I would suggest saying psychosocial factors instead of depression and anxiety and also add genetic and epigenetic factors in there too.

Introduction, line 86: Instead of "hunt for" I suggest "identify" instead.

Introduction, line 96: Here I would suggest finishing the sentence after citing (8, 9) and then start a new sentence to the effect, "As a result, there is a lack of consensus as to what dietary type is superior to produce weight loss (8, 10)".

Introduction, lines 106-111: I would suggest removing this section as the introduction is already very long.

Introduction, line 117: I would suggest using another term other than forgotten dark matter when referring to the gut microbiota.

Introduction, line 133: These studies only suggest that gut microbiota play a role in obesity pathogenesis and not outcome of LCD intervention so please remove the latter statement. The authors either need to find studies on the role of gut microbiota on weight loss after LCD or after another weight loss intervention such as bariatric surgery.

Results, line 179: It would help if the authors provide a statement on the reliability of three-day 24-hr dietary recall.

Results, line 232: Please provide citation on the previous use of the algorithm.

Results, line 250: Does the change in abundances in each of these three identified bacteria to be increased after LCD correlate with weight loss?

Results, line 256: It would help if the authors clarified at this stage how the two subgroups were defined for each group. Was this based on median values?

Results, line 317: Please provide citation for use of ANN.

Results, Figure 5: Please add in the caption title "after LCD" and also clarify in the caption itself which group of patients was analyzed.

Reviewer #2 (Comments for the Author):

The manuscript by Zhang, et. al., describes the effects of a short-term carbohydrate restricted diet on patients with obesity, with a specific focus on understanding the effects of GI microbial compositional. The manuscript is generally well written and easy to follow. However, I have a few suggestions and questions.

1. The major conclusion that the relative abundance of Bacteroidaceae Bacteroides is a positive predictor of outcomes is very interesting.

Given the limitations of 16s data for accurately predicting bacterial abundances (genomic copy number of the 16S gene varies between species) the authors should consider performing qPCR for Bacteroides to determine if absolute abundance confirms the 16s sequencing data.

2. More information of the statistical methods should be included in the methods, results and figure legends.

Specifically, many different comparisons are made for each data set but no indication of how or if correction for multiple comparisons is provided. In addition, with the repeated sampling there is no mentioned for how you accounted for repeated measures.

For example, in figure 2, its indicated that several bacterial species are more abundant in LCD group post treatment via unpaired, two-sided students t-test. If you only want to compare the baseline to the end of study for each group independently at the vary least this needs to be a paired t-test with correction for multiple comparisons, as the data is a repeated measure and more than one bacterial genera is being evaluated. However, given you actually have two groups with two sampling times, a repeated measures 2-way ANOVA, with corrections for multiple comparisons is more appropriate.

This holds for all of the other analysis in the figures.

3. Minor point, but the introduction is really long. This can easily be shortened.

Staff Comments:

Preparing Revision Guidelines

- Point-by-point responses to the issues raised by the reviewers in a file named "Response to

Reviewers," NOT IN YOUR COVER LETTER.

- Upload a compare copy of the manuscript (without figures) as a "Marked-Up Manuscript" file.
- Each figure must be uploaded as a separate file, and any multipanel figures must be assembled into one file.
- Manuscript: A .DOC version of the revised manuscript
- Figures: Editable, high-resolution, individual figure files are required at revision, TIFF or EPS files are preferred

For complete guidelines on revision requirements, please see the Instructions to Authors at [link to page]. **Submissions of a paper that does not conform to Microbiology Spectrum guidelines will delay acceptance of your manuscript.**

Please return the manuscript within 60 days; if you cannot complete the modification within this time period, please contact me. If you do not wish to modify the manuscript and prefer to submit it to another journal, please notify me of your decision immediately so that the manuscript may be formally withdrawn from consideration by Microbiology Spectrum.

If you would like to submit an image for consideration as the Featured Image for an issue, please contact Spectrum staff.

Manuscript #: Spectrum00223-21

Title: Gut microbiota serves a predictable outcome of short-term low-carbohydrate diet (LCD) intervention for patients with obesity

Susu Zhang^{1,2,#}, Peili Wu^{1,4,#}, Ye Tian^{1,2, #}, Bingdong Liu^{2,3,#}, Liuqing Huang², Zhihong Liu², Nie Lin^{1,5}, Ningning Xu¹, Yuting Ruan¹, Zhen Zhang¹, Ming Wang⁶, Zongbing Cui², HongWei Zhou⁷, Liwei Xie^{1,2,8,*}, Hong Chen^{1,*}, Jia Sun^{1,*}

Reviewer comments:

Reviewer #1:

1. Zhang et al aim to identify if differences in gut microbiota composition account for the variable outcomes of LCD in a group 51 overweight/obese Chinese patients. They find that LCD (10-25% calories from carbohydrates according to NLA guidelines) for 12 weeks resulted in greater changes in BMI and other adiposity indices than normal diet associated with reduced overall energy intake. This was independent of any major changes in fecal microbiota. The authors then show using a machine learning algorithm that 3 distinct gut microbiota species that produce butyrate are increased in the LCD group after intervention. Additionally, they find that Bacteroidaceae Bacteroides abundance at baseline is higher in the distinct weight loss LCD subgroup compared with the moderate weight loss subgroup and that this had greater predictive power using ANN analysis. This is an excellent study that is very well presented and executed and provides significant new insight into why the efficacy of LCD varies between people. My only real concern is the difference in BW at baseline between groups which needs to be mentioned in the discussion as a limitation.

Response: We thank Reviewer 1 for your assessment on our manuscript. Yes, indeed, participants for RCT study were randomly selected and assigned into both groups. Upon completion of the study, we noticed the average body weight for LCD

group was heavier than that for ND group at the baseline (BMI for ND and LCD: 28.61 ± 2.04 vs. 30.44 ± 3.38 kg m⁻²), which is a disadvantage of this study, but is difficult to avoid. This discrepancy has been discussed in the section of limitations. (Page22, line 498-514)

2. There is also a little ambiguity about which class of bacteria could be contributing more to the outcomes of LCD. Is it the increase in SCFA-producing bacteria during intervention or is it the high levels of Bacteroidaceae Bacteroides at baseline or is it both? Maybe a schematic diagram could help clarify this issue. Further from this point, while the role of SCFA including butyrate in regulating energy balance is discussed, what factors produced by Bacteroidaceae Bacteroides could contribute to the outcome of LCD? Finally, if the authors cannot measure SCFA in fecal samples, this needs to be mentioned as another limitation.

Response: Thank you very much for your comments. We believe there are fundamental issues with our interpretations as we are classifying SCFA-producing bacteria either between ND and LCD or between LCD_MG and LCD_DG to be analyzed and discussed. In figure 2D-F, we took advantage of robust statistical analysis, e.g. random forest to identify that *Porphyromonadaceae Parabacteroides*, *Odoribacteraceae Butyricimonas*, and *Ruminococcaceae Oscillospira* are critical bacteria members between ND and LCD group after intervention. They were not significantly different at baseline, but robustly increased after LCD intervention. Based on the existing literature, these bacteria are all linked to the production of beneficial SCFAs, such as butyrate in the GI track. Meanwhile, butyrate also could stimulate the production of gut hormones (e.g., glucagon-like peptide-1, GLP-1) and decrease food intake to alleviate obesity (1) . We assume participants in LCD group could benefit from these bacteria and their metabolites upon low carbohydrate diet intervention. These findings will provide fundamental basis for our upcoming clinical studies of low carbohydrate and/or probiotics intervention.

In the subgroups of LCD intervention between LCD_MG and LCD_DG, these

participants are further divided into moderate and distinct weight loss group based on their weight loss efficacy. Same as above, random forest is utilized to select the critical bacteria between two subgroups. *Bacteroidaceae Bacteroides* is the top listed candidate with highest relative abundance (Figure4G-I). Although previous studies demonstrated that *Bacteroidetes* is the largest propionate producers in the human gut(2, 3) and its level correlates with fecal levels of SCFAs (4), its beneficial effect upon increased relative abundance in gut is confirmed via stimulating overall SCFAs production in gut. SCFAs are derived from intestinal microbial fermentation of indigestible foods and are the main energy source of colonocytes, making them crucial to gastrointestinal health(5). Our work demonstrated that a higher relative abundance of *Bacteroidaceae Bacteroides* at baseline was significantly associated with superior weight loss after a short-term LCD intervention. However, in present investigation, one of the major limitation is that we failed to conduct fecal metabonomic detection due to the exhausted samples, the content of butyrate or propionate in the intestine of the participants could not be determined yet. We have added this to be one of limitations in manuscripts. (Page 22, line 498-514)

Reference

1. Coppola S, Avagliano C, Calignano A, Berni Canani R. 2021. The Protective Role of Butyrate against Obesity and Obesity-Related Diseases. *Molecules* 26:682.
2. Aguirre M, Eck A, Koenen ME, Savelkoul PHM, Budding AE, Venema K. 2016. Diet drives quick changes in the metabolic activity and composition of human gut microbiota in a validated in vitro gut model. *Res Microbiol* 167:114–125.
3. Salonen A, Lahti L, Salojärvi J, Holtrop G, Korpela K, Duncan SH, Date P, Farquharson F, Johnstone AM, Lobley GE, Louis P, Flint HJ, De Vos WM. 2014. Impact of diet and individual variation on intestinal microbiota composition and fermentation products in obese men. *ISME J* 8:2218–2230.
4. Zhao Y, Wu J, Li J V., Zhou NY, Tang H, Wang Y. 2013. Gut microbiota composition modifies fecal metabolic profiles in mice. *J Proteome Res*

12:2987–2999.

5. Canfora EE, Jocken JW, Blaak EE. 2015. Short-chain fatty acids in control of body weight and insulin sensitivity. *Nat Rev Endocrinol* 11:577–591.

3. Introduction, line 70: As causes of obesity, I would suggest saying psychosocial factors instead of depression and anxiety and also add genetic and epigenetic factors in there too.

Response: Thanks for the suggestion. Multiple factors contribute to obesity, depression or anxiety is one of psychosocial factors. we have changed it to “**psychosocial factors**”. (Page 3, line70)

4. Introduction, line 86: Instead of "hunt for" I suggest "identify" instead.

Response: Thank you very much for your suggestion, we have changed “**hunt for**” into “**identify**”. (Page 4, line 80)

5. Introduction, line 96: Here I would suggest finishing the sentence after citing (8, 9) and then start a new sentence to the effect, "As a result, there is a lack of consensus as to what dietary type is superior to produce weight loss (8, 10)".

Response: Thanks for your advice. We have revised the sentence to: “**These eating patterns with varying macronutrient distributions have substantial/spurious benefits in certain groups of patients (8, 9). As a result, there is a lack of consensus as to what dietary type is superior to produce weight loss (8, 10).**” (Page 4, line 88-91)

6. Introduction, lines 106-111: I would suggest removing this section as the introduction is already very long.

Response: Thank you very much for your suggestion. This part has been removed in revised manuscript.

7. Introduction, line 117: I would suggest using another term other than forgotten

dark matter when referring to the gut microbiota.

Response: Thank you very much for your suggestion. As the gut microbiota live in gastrointestinal track, we changed the “**forgotten dark matter**” into “**gut microbiota**”. (Page 5, line 106)

8. Introduction, line 133: These studies only suggest that gut microbiota play role in obesity pathogenesis and not outcome of LCD intervention so please remove the latter statement. The authors either need to find studies on the role of gut microbiota on weight loss after LCD or after another weight loss intervention such as bariatric surgery.

Response: Thank you very much for your comments. In this part, we added citation and adjusted the sentence as “**Fouladi F. et.al. reported that bariatric surgery such as Roux-en-Y Gastric bypass (RYGB) surgery significant shifts in the gut microbiota which could potentially contribute to weight loss and metabolic benefits. Together, these results implied that Bacteroides and Firmicutes may play a diverse role in the pathogenesis of obesity.**” (Page 5, line 119-122)

9. Results, line 179: It would help if the authors provide a statement on the reliability of three-day 24-hr dietary recall.

Response: Thank you very much for the suggestion. 24-hr dietary recall is a dietary assessment tool, 3-day 24-hr dietary recalls, which are used by medical professionals, nutrition specialists, and social scientists. Statement and citation have been added (Page 8, line168-169).

10. Results, line 232: Please provide citation on the previous use of the algorithm.

Response: Thank you very much for your suggestion. We have added citation for random forest algorithm used in our study (Page 10, line 223).

11. Results, line 250: Does the change in abundances in each of these three identified

bacteria to be increased after LCD correlate with weight loss?

Response: Thank you very much for your comments. Here, we take advantage of Mantel test to measure correlation between two matrices. Gut microbiota is considered as a whole part not an individual. Thus, we need to calculate the correlation between two matrices, gut microbiota and clinical parameters (BMI, waist circumference, WHR, BFR and VFA), as two matrices. For three key gut microbiota, we built a matrix, containing the relative abundance the change (**Cluster**) in abundance after LCD intervention. Here, in Mantel test, this Cluster representing the change of relative abundance positively correlates with clinical parameters such as BMI, waist and BFR ($p < 0.05$). (**Page 11 line 237-239**)

12. Results, line 256: It would help if the authors clarified at this stage how the two subgroups were defined for each group. Was this based on median values?

Response: Thank you very much for your comments. The efficacy of LCD intervention varies individually. To identify the confounders that led to this variation, participants were divided into two subgroups: moderate (MG) and the distinct (DG) weight loss group. These two subgroups were defined based on cluster stratification. Here, we add justification for two-group separation “**Further analysis of weight loss**

outcome on changes of BMI, waist circumference, WHR, BFR and VFA for each participant through cluster stratification, the median of these five anthropometric parameters was taken as the critical cutting-point.” (Page 11, line 245-248)

13. Results, line 317: Please provide citation for use of ANN.

Response: Thank you very much for your suggestion. Three citations were added plus description of its implication in biomedical research as **“In recent years, an increasing number of medical studies have applied ANN model to process complex data because of its superiority, such as to seek predictors of catheter-related thrombosis in hospitalized infants (37), to calibrate the prediction of survival in glioblastoma patients (38), and to optimize the auxiliary diagnosis of insomnia disorder (39).” (Page14, line 309-313)**

14. Results, Figure 5: Please add in the caption title "after LCD" and also clarify in the caption itself which group of patients was analyzed.

Response: Thanks for the advice. We have added **“after LCD”** in the caption title of Figure 5. Samples from participants of **LCD group** were analyzed which had been clarified. **(Page 46, line 995-998)**

Reviewer #2:

The manuscript by Zhang, et. al., describes the effects of a short-term carbohydrate restricted diet on patients with obesity, with a specific focus on understanding the effects of GI microbial compositional. The manuscript is generally well written and easy to follow. However, I have a few suggestions and questions.

1. Given the limitations of 16s data for accurately predicting bacterial abundances (genomic copy number of the 16S gene varies between species) the authors should consider performing qPCR for Bacteroides to determine if absolute abundance confirms the 16s sequencing data.

Response: We thank Reviewer 2 for your effort to assess our work. We believe

that lacking of qPCR to determine the absolute abundance of Bacteroides could be a fundamental issue. However, we ran out of fecal samples collected for this LCD-based weight loss clinical intervention. This was a limitation of this study, that has been discussed in the limitation section. (Page 22, line 498-514).

2. More information of the statistical methods should be included in the methods, results and figure legends. Specifically, many different comparisons are made for each data set but no indication of how or if correction for multiple comparisons is provided. In addition, with the repeated sampling there is no mentioned for how you accounted for repeated measures. For example, in figure 2, its indicated that several bacterial species are more abundant in LCD group post treatment via unpaired, two-sided students t-test. If you only want to compare the baseline to the end of study for each group independently at the vary least this needs to be a paired t-test with correction for multiple comparisons, as the data is a repeated measure and more than one bacterial genera is being evaluated. However, given you actually have two groups with two sampling times, a repeated measures 2-way ANOVA, with corrections for multiple comparisons is more appropriate. This holds for all of the other analysis in the figures.

Response: Thank you very much for your comments. In both method and figure legend section, we have revised the description of statistical method used for each figure. Updated information was added and highlight in Red. (Page 29 line 641-645; Page 42 line943-945; Page 44 line984-986).

Regarding the paired analysis before and after LCD intervention, we actually want to compare the difference of relative abundance of each key microbiota. We agree with your comments that a repeated measure 2-way is more appropriate than the unpaired student t-test. For this aim, we performed two-way ANOVA with repeated measurement followed by a Tukey post hoc test for **Figure 2F, Figure 4I and Figure S5**.

As a result, all analyses except for **Figure 2F** were kept consistent. We repeated 2-ways ANOVA for Figure 2F. The relative abundance of *Odoribacteraceae*

Butyricimonas was higher at end stage, but the *P* value > 0.05. Figure 2F has been updated in the revised manuscript. We also made corresponding change on manuscript “**More specifically, the relative abundance of *Ruminococcaceae Oscillospira* was higher comparing to the baseline. Meanwhile, the relative abundance of *Odoribacteraceae Butyricimonas* had an increasing trend but did not reach statistical difference after 12-week LCD intervention. Other than these, another bacterial biomarker was identified *Porphyromonadaceae Parabacteroides* also had higher relative abundance after 12 weeks of LCD intervention**” (Page11, line231-237). Therefore, we deleted sentences related to *Odoribacteraceae Butyricimonas* in discussion sections.

The statistical methods had been updated in “Other Statistical Analysis” of “MATERIALS AND METHODS” section as well as figure legends.

3. Minor point, but the introduction is really long. This can easily be shortened.

Response: Thank you very much for the suggestion. We have revised the introduction section into three pages.

August 4, 2021

Dr. Liwei Xie

Guangdong Provincial Key Laboratory of Microbial Culture Collection and Application, State Key Laboratory of Applied Microbiology Southern China, Institute of Microbiology, Guangdong Academy of Sciences, Guangzhou, 510070, China
Guangzhou, Guangdong 510070
China

Re: Spectrum00223-21R1 (Gut microbiota serves a predictable outcome of short-term low-carbohydrate diet (LCD) intervention for patients with obesity)

Dear Dr. Liwei Xie:

Thank you for responding to the Reviewer's queries. To meet the Journal's requirements on data availability, please deposit the sequencing data in one of the listed repositories: <https://journals.asm.org/list-data-repositories> and note this in the manuscript file.

Thank you for submitting your manuscript to Microbiology Spectrum. As you will see your paper is very close to acceptance. Please modify the manuscript along the lines I have recommended. As these revisions are quite minor, I expect that you should be able to turn in the revised paper in less than 30 days, if not sooner. If your manuscript was reviewed, you will find the reviewers' comments below.

When submitting the revised version of your paper, please provide (1) point-by-point responses to the issues I raised in your cover letter, and (2) a PDF file that indicates the changes from the original submission (by highlighting or underlining the changes) as file type "Marked Up Manuscript - For Review Only". Please use this link to submit your revised manuscript. Detailed information on submitting your revised paper are below.

Link Not Available

Sincerely,

Steven Frese

Reviewer comments:

Preparing Revision Guidelines

- point-by-point responses to the issues I raised in your cover letter
- Upload a compare copy of the manuscript (without figures) as a "Marked-Up Manuscript" file.
- Each figure must be uploaded as a separate file, and any multipanel figures must be assembled into one file.
- Manuscript: A .DOC version of the revised manuscript
- Figures: Editable, high-resolution, individual figure files are required at revision, TIFF or EPS files are preferred

For complete guidelines on revision requirements, please see the Instructions to Authors at [link to page]. **Submissions of a paper that does not conform to Microbiology Spectrum guidelines will delay acceptance of your manuscript.**

Please return the manuscript within 60 days; if you cannot complete the modification within this time period, please contact me. If you do not wish to modify the manuscript and prefer to submit it to another journal, please notify me of your decision immediately so that the manuscript may be formally withdrawn from consideration by Microbiology Spectrum.

If you would like to submit an image for consideration as the Featured Image for an issue, please contact Spectrum staff.

Manuscript #: Spectrum00223-21

Title: Gut microbiota serves a predictable outcome of short-term low-carbohydrate diet (LCD) intervention for patients with obesity

Susu Zhang^{1,2,#}, Peili Wu^{1,4,#}, Ye Tian^{1,2, #}, Bingdong Liu^{2,3,#}, Liuqing Huang², Zhihong Liu², Nie Lin^{1,5}, Ningning Xu¹, Yuting Ruan¹, Zhen Zhang¹, Ming Wang⁶, Zongbing Cui², HongWei Zhou⁷, Liwei Xie^{1,2,8,*}, Hong Chen^{1,*}, Jia Sun^{1,*}

Thank you for responding to the Reviewer's queries. To meet the Journal's requirements on data availability, please deposit the sequencing data in one of the listed repositories: <https://journals.asm.org/list-data-repositories> and note this in the manuscript file.

RESPONSE: Thank you very much for your comment. We have upload 16S rDNA sequencing data file to NCBI, with project# PRJNA752174, with the link at <https://www.ncbi.nlm.nih.gov/bioproject/PRJNA752174>.

August 15, 2021

Dr. Liwei Xie

Guangdong Provincial Key Laboratory of Microbial Culture Collection and Application, State Key Laboratory of Applied Microbiology Southern China, Institute of Microbiology, Guangdong Academy of Sciences, Guangzhou, 510070, China
Guangzhou, Guangdong 510070
China

Re: Spectrum00223-21R2 (Gut microbiota serves a predictable outcome of short-term low-carbohydrate diet (LCD) intervention for patients with obesity)

Dear Dr. Liwei Xie:

Your manuscript has been accepted, and I am forwarding it to the ASM Journals Department for publication. You will be notified when your proofs are ready to be viewed.

Sincerely,

Steven Frese
Editor, Microbiology Spectrum

Journals Department
Table S1: Accept
Supplementary Figure: Accept